# VISTA, PDL-L1, and *BRAF*—A Review of New and Old Markers in the Prognosis of Melanoma

**DOI:** 10.3390/medicina58010074

**Published:** 2022-01-04

**Authors:** Andreea Cătălina Tinca, Iuliu Gabriel Cocuz, Mihaela Cornelia Șincu, Raluca Niculescu, Adrian Horațiu Sabău, Diana Maria Chiorean, Andreea Raluca Szőke, Ovidiu Simion Cotoi

**Affiliations:** 1Doctoral School of Medicine and Pharmacy, University of Medicine, Pharmacy, Sciences and Technology “George Emil Palade” of Targu Mures, 540142 Targu Mures, Romania; iuliu.cocuz@umfst.ro (I.G.C.); mihaela.sincu02@gmail.com (M.C.Ș.); niculescuralu@yahoo.com (R.N.); sabauhoratiu@gmail.com (A.H.S.); chioreandianamaria@yahoo.com (D.M.C.); szoke.andreea@yahoo.com (A.R.S.); 2Pathology Department, Mures Clinical County Hospital, 540011 Targu Mures, Romania; 3Pathophysiology Department, University of Medicine, Pharmacy, Sciences and Technology “George Emil Palade” of Targu Mures, 540142 Targu Mures, Romania

**Keywords:** melanoma, VISTA, PD-L1, *BRAF*, prognosis

## Abstract

Melanoma is currently known as one of the most aggressive malignant tumors. The prognostic factors and particularities of this neoplasm are a persistent hot topic in the medical field. This review has multiple purposes. First, we aim to summarize the known data regarding the histological and immunohistochemical appearance of this versatile tumor and to look further into the analysis of several widely used prognostic markers, such as B-Raf proto-oncogene, serine/threonine kinase *BRAF*. The second purpose is to analyze the data on the new prognostic markers, V-domain Immunoglobulin Suppressor of T cell Activation (VISTA) and Programmed death-ligand 1 (PD-L1). VISTA is a novel target that is considered to be highly important in determining the invasive potential and treatment response of a melanoma, and there are currently only a limited number of studies describing its role. PD-L1 is a marker with whose importance has been revealed in multiple types of malignancies, but its exact role regarding melanoma remains under investigation. In conclusion, the gathered data highlights the importance of correlations between these markers toward providing patients with a better outcome.

## 1. Introduction

Melanoma is a malignant tumor originating in melanocytic (melanin-producing) cells that has an increased potential for invasion and metastasis; therefore, its early diagnosis and treatment are very important for improving the prognosis of patients. Since melanoma is one of the most aggressive tumors known and the worldwide incidence of this neoplasia is increasing from year to year, the aggressiveness of the tumor and its implications for the lives of patients ensure that it remains a very important topic.

From the point of view of classification according to the involved site, there are multiple types of melanoma, from which we are going to mention the following: cutaneous melanoma (which can be related with sun exposure and also arise in sun-shielded sites), mucosal melanoma (genial, oral, sinonasal), ocular melanoma, and melanoma arising in congenital naevus. The objectives of this review take into consideration the various forms of cutaneous melanoma related to sun exposure [1].

During embryonic life, melanocytes develop from the neural crest. At first, cells constitute a group of ectodermal cells, which originate from the external layer of the neural tube. In the next stages of embryo differentiation, and then during the formation of the fetus, these ectodermal cells migrate throughout the entire body and continue to differentiate into different components.

The precursor cells of melanocytes are called melanoblasts. These cells are derived from melanocytic precursors and will proliferate as they migrate toward the tegument.

The malignant transformation of melanocytes is the result of interactions between components related to the individual and the environment, elements that are generally found in the genesis of any type of cancer. Among the factors belonging to the environment, we highlight ultraviolet (UV) radiation, which is considered the most important risk factor in the appearance of cutaneous melanoma. Genetic factors, as well as family history, are also considered to be important risk factors in melanoma genesis. The genes most heavily linked to the appearance of this cancer, according to the World Health Organization, are *BRAF* and *KRAS* [2,3].

### 1.1. Characteristics

In the presence of appropriate risk factors, melanoma can occur at any level of the skin. This tumor can arise de novo or originate from a melanocytic nevus. According to World Health Organization (WHO) guidelines, about 1 in 33,000 nevi will begin developing into melanoma. It is considered that cutaneous melanoma is responsible for more than 90% of the mortality involving all skin pathologies.

The gold standard for the diagnosis of melanoma is represented by the histological and immunohistochemical methods that are currently used in pathology laboratories, as well as new methods that are related to the genomic analysis of malignant tumors [1,4].

According to McGovern and Clark, there are two main stages in the development and progression of melanoma. The first is illustrated by the appearance of a pigmented area that extends radially (on the surface of the skin, along a horizontal axis). The second stage is characterized by growth that manifests the formation of a nodule [1,4,5].

Patients that present melanoma metastases have an average survival of about 8 months from the moment in which the first metastasis has been confirmed. Among the criteria that ensures the prognosis of the tumor, we mention the correlation of the clinical aspects related to the patient (age, comorbidities) with the morphological aspects of the tumor, such as the Clark level and its thickness according to the Breslow index. According to the WHO, Clark level has five components: (1) melanoma in situ limited to the epidermis, (2) papillary dermis invasion, (3) papillary-reticular dermis junction invasion, (4) reticular dermis invasion, and (5) hypodermic invasion. As for the Breslow index (which represents the thickness of the tumor, measured from the top of the granular layer of the epidermis or from the base of the ulcer, if the surface is ulcerated, to the deepest invasive cell across the base of the tumor), higher values are associated with worse prognosis and increased risk of sentinel node metastasis [1,4,6].

### 1.2. Classification 

The World Health Organization (WHO) classifies cutaneous melanoma from the perspective of its association with ultraviolet exposure, called cumulative solar damage. In this category, we mention the types of melanoma associated with exposure to sunlight (such as superficial melanoma, lentigo maligna, desmoplastic melanoma). The second category includes the types of melanoma that are not considered to be associated with UV exposure and their effects on the skin (acral melanoma, melanoma of the mucous membranes, uveal melanoma). A special category is nodular melanoma, a tumor that occurs regardless of the way in which the malignant transformation of melanocytes takes place and, therefore, this type is found in and described for both classifications. The synonyms of the tumor are, according to WHO, “rapidly growing melanoma” and “primary melanoma without a radial phase”. A possible explanation for this occurrence is that nodular melanoma presents the genetic alterations required for the tumor to progress rapidly, so the barriers that keep the tumor from deep invasion can be overrun early. Histologically, nodular melanoma presents as a tumor located in the dermis, with nests of tumoral cells that are extended towards the epidermis and cause ulceration. Pagetoid pattern is uncommon. The cells composing the tumor are most often epitheliod, either uniform or varied. Intratumoral lymphocytic infiltration is a common feature and it is more often encountered when the tumoral cells present different morphology [1,4].

## 2. Inflammatory Cells—Tumor Infiltrating Lymphocytes

Inflammatory cells from the tumoral microenvironment present in primary melanocytic malignancies are always reported under the name of ‘brisk’ or ‘non-brisk’. The brisk infiltrate is defined by lymphocytes that are present in the entire invasive component of the tumor diffusely, or simply described as lymphocytes infiltrating across the entire base of the vertical growth phase. The non-brisk infiltrate is defined by lymphocytes that are distributed focally and it is not found along the base of the invasive component. Inflammatory infiltrate has an important prognostic significance. Its role was especially highlighted after the development and implementation of immunological therapy, demonstrating that these inflammatory cells not only determine the prognosis of survival for patients, but also of cancer therapy. In the last few years, the prognostic value of tumor infiltrating lymphocytes (TILs) has been debated in numerous studies. A particular area of interest is the peritumoral inflammatory infiltrate, which is not as well studied as TILs. Recent studies have mentioned that the presence of cytotoxic lymphocytes (CD8+) is associated with a prognostic role in immunotherapy, being more relevant for the patient compared to the intratumoral inflammatory infiltrate represented by the same type of lymphocytes, as well as CD4-+ and PD-1-+ lymphocytes and PD-L1-+ cells [7]. During the early research period, reports quantified the score of TILs on hematoxylin/eosin sections. Inflammatory infiltrate is considered to be an independent prognostic factor. Regarding brisk, it was highlighted that it is associated with better survival compared to non-brisk infiltrate. There are studies that contradict these data, such as the one conducted by Rao et al., which argues that the difference in survival between cases with brisk and non-brisk inflammatory infiltrate is not statistically significant. However, it also demonstrated better survival in patients with a high TILs score compared to the group of patients who did not show inflammatory infiltrate [8]. A study led by Eriksson et al. did not confirm the TILs score as an independent prognostic factor [9]. It is possible that the results of these various studies are based on specific particularities of each case, such as the stage and type of melanoma involved. Immunohistochemical studies were also performed to reveal the types of inflammatory cells present in the tumor, showing that CD69 and CD20 lymphocytes—as well as cytotoxic T lymphocytes—are associated with much better prognosis [9,10].

Regarding sentinel nodules, the TILs score is considered to be inversely proportional to the risk of metastasis when using classifications described by both Clark and the Australian Melanoma Institute [10,11].

However, most studies were focused on primary tumors, rather than secondary and metastatic tumors. A study conducted by Kakavand and Wong has shown that the presence of TILs in metastatic tumors (inflammatory cells being identified using immunohistochemistry techniques) are independent prognostic factors [12,13].

Thus, regarding the inflammatory infiltrate and its role in melanoma, we conclude that both intratumoral and peritumoral inflammatory infiltrate are important prognostic factors in the therapy against this malignant tumor.

## 3. Immunohistochemistry

The use of immunohistochemistry in the diagnosis of melanoma is known and acknowledged worldwide. This analysis is an important tool which provides not only the diagnosis, but also can orientate the patients towards a certain prognosis. Immunohistochemistry staining has increased significantly in recent years, not only for melanoma, but for all types of neoplasms. A study conducted by Dinehart showed that a majority (95%) of surveyed dermatopathologists are using immunohistochemistry (IHC) in their practice [14]. Other studies highlight the increase in IHC use in up to 25% of melanoma cases. IHC analysis is shown to be highly valuable especially in cases of poorly differentiated neoplasms [15].

In addition to assessing lineage, IHC is extremely important in cases in which melanoma shows regression or fibrotic changes, especially when pigmentation is absent. The same review described the utility of IHC mainly in cases of melanoma in situ and it was used mostly in order to assess the possibility of invasion in these cases [15,16,17].

## 4. VISTA

The concept of immunological therapy in malignant neoplasia was mentioned and proposed for practice by Burnet, starting from the fact that the immune system has a role in detecting neoplastic cells and removing them from the body. VISTA (V domain Ig containing suppressor of T-cell activation) is considered a new therapeutic target for anticancer therapy. This molecule is part of the B7 family associated with checkpoint receptors and is the counterpart of PD-1 and PD-L1, also having ligand activity on APC and T lymphocytes or, as some research claims, has a suppressive effect on the mentioned inflammatory cells [18].

VISTA currently has a controversial role, the modulation of the tumor microenvironment remains an unsolved subject. Targeted studies on the molecule have shown that it is predominantly expressed by hematopoietic cells, especially granulocytes, but also by other cells of the myeloid line, the expression being lower than T lymphocytes. More precisely, VISTA is expressed by immature and mature myeloid line cells, CD4+ T lymphocytes, and CD8+ T lymphocytes. There are studies that have revealed that therapy with anti-VISTA agents have blocked the acute graft rejection against the host and had immunosuppressive effect due to their ability to remove cells that express VISTA [18,19,20].

The potential of VISTA in oncotherapy has been described and demonstrated for the first time in a case of a patient with fibrosarcoma. After a number of studies made on murine models for cancer, VISTA was exclusively identified on intratumoral leukocytes. A very important aspect about VISTA that is highlighted in the research up to now suggests the efficiency of VISTA blocking therapy, including in the presence of PD-L1. Therefore, VISTA and PD-L1 are considered to be independent pathways, and numerous important benefits can be obtained when both are targeted together in immunotherapy [20,21].

Studies in melanoma patients have shown that VISTA is associated with dysfunctions of T lymphocytes in this pathology. One of these studies, led by Resenbaum, that is conducted on patients with stage III cutaneous melanoma explains that intratumoral inflammatory infiltrate—also known as TILs—is important for the better prognosis of patients. Thus, the cases evaluated in that study were reported as having either increased or low expression of VISTA and were correspondingly evaluated in terms of survival [22].

Among the results obtained in this study, we observe the decrease in survival in patients in the category with low expression of VISTA (approximately 4 years after diagnosis) compared to the cohort of patients with high expression (survival up to 10 years after diagnosis). Another part of the study concerned melanoma samples collected from surgery. The inflammatory cells present at the level of the tumor showed different levels of VISTA and were marked with the immunohistochemical marker CD45. These tumors received a score, called an L-score, which quantified the density but also the distribution of the lymphocyte population in each case of melanoma. VISTA has been identified using cytometry, and elevated levels are correlated with an increased number of lymphocytes, as well as in cases where the number of inflammatory cells has decreased, raising suspicion that this immunoglobulin is not exclusively expressed by cells of the immune system [22,23].

Specific immunohistochemical reactions are performed to identify the presence of VISTA in melanocytic tumor cells. In such cases, no significant differences were observed in terms of age, gender, mutant status of patients, or the stage or tumor location. These findings are extremely important, because it shows that the tumoral cells, and not only the inflammatory cells, can express this marker [23,24].

Another study was conducted by Lawrence, who wanted to analyze the expression of VISTA in melanoma, with the aim of comparing its expression with PD-L1 and the association between this molecule and the prognosis of patients [25]. A total of 85 studies that considered the identification of PD-L1, CD3, and VISTA by immunohistochemical determinations were selected. Intratumoral granulocytes were identified based on their morphology. The density of the intratumoral inflammatory infiltrate was quantified with the help of CD3 immunostaining and by following the morphological appearance in hematoxylin/eosin-stained sections. In this study, VISTA was negative in melanocytic tumor cells, and most of the expression was revealed in neutrophils in cases where ulcerated tumors presented. Weaker expression of the molecule has been identified in mononucleated cells, including in lymphocytes, macrophages, and monocytes [25,26].

Overexpression of VISTA inhibits the immune system by suppressing the proliferation of T lymphocytes and cytokine secretion (for example, IL-10, TNF alpha, interferon gamma). VISTA blocking can be used as an immunomodulator such that tumor growth will be slowed down, a fact that has been demonstrated using experimental murine models. The use of molecular methods in experimental studies has shown that VISTA is expressed exclusively in hematopoietic cells, contrary to other studies that have been carried out, which have described its expression also in tumoral cells. In the samples evaluated from mice, but also of tested human tissues, this aspect was confirmed [27,28].

Eventually, a review of VISTA on multiple types of cancer, conducted by Huang, revealed the heterogeneity in expression that this marker can have. In some cases, VISTA was associated with a worse prognosis, while in others, it was shown to be expressed in early tumoral stages or certain cancer subtypes. Overall, either high or low levels have been observed, and levels were positively correlated with the presence of immune cells, further highlighting the complex appearance and effect of VISTA [29,30].

## 5. PD-L1

PD-1 is a molecule that was first discovered and described in 1992 as being associated with diseases such as glomerulonephritis or splenomegaly and having a role in cell apoptosis. In addition to these findings, it has been stated that PD-1 also has a role in gastritis and dilated cardiomyopathy, as demonstrated by experimental studies in mice. Following research conducted by Nishimura et al. over the years, it was concluded that PD-1 is a molecule that limits both the activation and proliferation of T lymphocytes, thus promoting tolerance to itself. PD-1 ligand, the molecule known as PD-L1, has been identified as a homodimer molecule B7-1/B7-2. PD-L1 is a glycoprotein type 1 that is expressed by different cells compared to PD-1. First of all, PD-1 is found in activated B and T lymphocytes, especially CD8+ cytotoxic lymphocytes, but also in dendritic cells. PD-L1 is found in epithelial, endothelial, and T lymphocytes, and is upregulated in several types of tumors as a mechanism of shunting the immune system. To date, numerous tumors have been identified that present PD-1 and PD-L1, including tumors that are both solid and those that have developed in hematogenous lines. Currently, therapies against PD-1 and its ligand have been widely applied around the world for treatment of some types of cancer and appear to be promising for even more types of malignancies [31,32].

The role of PD-1 and its ligand has also been studied in melanoma. Within the last decade, in 2014, the first drug was approved for immunotherapy against these molecules present in the tumor microenvironment. The results obtained from the new treatment (nivolumab) were significantly better compared to those who followed the classical therapy, whether chemotherapy or based on *BRAF* inhibitors. For example, in patients who received immunotherapy, the response rate was 31.7% compared to 10.6% of those undergoing chemotherapy [33]. In 2017, 3 years after anti PD-1/PD-L1 therapy started to be used, the Food and Drug Administration (FDA) approved the use of nivolumab—including in patients with fully resectioned melanoma—but that, however, had distant metastases or lymphonodular metastases, thus targeting patients in stages IIIB, IIIC, and IV. Compared to the existing therapy at that time, immunotherapy proved to be more effective in terms of prognosis, improving the survival rate by 13% [34,35,36].

Nivolumab is not the only drug designed against PD-1 and PD-L1. Another therapy is pembrolizumab, medication intended exclusively for PD-1, which is predominantly used in metastatic melanoma in cases where the primary tumor is unresectable [33,37].

There are a significant number of active clinical trials currently investigating the therapeutic potential of PD-1/PD-L1 inhibition, targeting not only melanoma but also ovarian, breast, and central nervous system tumors and some sarcomas. Progress has been made in understanding the role of the immune system and these pathways. For example, in breast cancer, PD-L1 expression was correlated with triple negative cases, where pembrolizumab had a lasting effect, and nivolumab was associated with a better prognosis, both alone and in combination therapies. Regarding ovarian cancer, PD-L1 is considered to be a positive prognostic factor, with therapy with both agents (pembrolizumab and nivolumab) demonstrating a considerable antitumor profile. If we refer to tumors of the central nervous system, we can state, according to the literature, that the PD-1/PD-L1 blockade could represent a new therapy in glioblastomas (extremely aggressive malignant tumors). For sarcoma, the mentioned medication was tested alone and the good tolerance of patients to pembrolizumab was demonstrated [38,39,40,41].

Combinations of PD1/PD-L1 inhibitors with targeted therapies and other inhibitors of the immune system are being researched. Studies on the response to chemotherapy have shown that this form of treatment can amplify the immune response against cancer by inducing PD-L1 expression in tumor cells but also by facilitating the response to immunotherapy. Such chemotherapy–immunotherapy combinations have been carried out in clinical trials that have targeted patients with lung cancer and melanoma. As a result, an increase in treatment response was identified in pembrolizumab, with a difference of 26% compared to chemotherapy administered as the only method of treatment [42,43].

The toxicity of PD-1/PD-L1 medication is considered to be lower and better accepted by patients compared to treatment with chemotherapy or other immunotherapeutic agents. A meta-analysis of randomized clinical trials in patients receiving pembrolizumab and nivolumab describes similar survival for both drugs and also similar risks of developing side effects such as tubulations of thyroid function (hyperfunction and hypofunction alike), different types of colitis, pruritus, and pneumonitis [44].

## 6. *BRAF*

*BRAF* mutations in melanoma are very well known and documented, being described to promote and sustain oncogenesis by inhibition of the apoptosis process and tumor suppressor inactivation. These mutations, especially *BRAF V600E*, are observed in approximately half of melanoma cases, and this specific mutation represents 90% of all known *BRAF* mutations [45]. Many studies have highlighted the fact that *BRAFV600E* is even more frequently encountered in melanomas that arise on a nevi. The activated mutation turns on the mitogen-activated protein kinase (MAPK) pathway. However, despite the fact that *BRAF* is very important in the development and progression of melanoma, studies have shown that it alone is not sufficient for tumor genesis [45,46]. Recent research studies previously mentioned, such as the one conducted by Rosenbaum, have attempted to compare the expression and the effect of *BRAF* in relation to the novel marker VISTA. It was confirmed that blockade of *BRAF* influences the levels of VISTA. Therefore, the combination of PLX4720 with the Mitogen-activated protein kinase kinase (MEK) inhibitor PD0325901 against *BRAF* lead to a decrease in VISTA expression. All the data collected from the study and Western blot analysis (used for the detection of VISTA) showed that *BRAF* inhibition upregulates *Forkhead Box D3*(*FOXD3*) (molecule involved in transcription processes in adult life) and suppresses the expression of VISTA [47,48,49].

## 7. Conclusions

Melanoma is a versatile tumor with a variety of parameters that need to be taken into consideration. Currently, the most important predictive factors are Clark invasion and Breslow index, yet there are multiple factors that can change the prognosis. One of them is *BRAF*, the most common gene involved in the pathogenesis of the tumor. PD-L1 is a novel factor described, for which new therapies are already promising. The newest factor targeted is VISTA. The exact role of VISTA in patients with melanoma (and cancer, more generally) requires further investigations. In the literature, so far, few studies have been published about the expression of this new marker, especially in the context of melanoma, and the collected data differ from study to study. Some studies correlated the presence of VISTA with low survival in cases in which the marker is highly expressed, with a better prognosis for patients that have low VISTA expression, while other studies contradict these findings. So far, most of the collected data show VISTA as being exclusively present in inflammatory cells; however, the expression in the tumoral melanocytes has also been recently identified, yet contradicted by ongoing research. The link between *BRAF* and VISTA has been highlighted in studies that followed the response of VISTA-expressed tumors in the anti-*BRAF* blockade, which showed a decreased expression of VISTA over time, especially in combined anti-*BRAF* therapy.

The relationship between VISTA and PD-L1 is also currently under investigation. VISTA and PD-L1 both target the immune system and the tumor microenvironment with differences in expression from case to case. It has been demonstrated that the tumors which express VISTA can still respond to PD-1/PDL-1 blockade, but the response is heterogeneous since the pathways of these molecules are different. Therefore, given the present findings regarding prognosis and the correlation with VISTA demonstrated to date, the blockade of both VISTA and PD-L1 should be a priority in the future research in patients with melanoma.

## Data Availability

All the data produced here is available and can produced when required.

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
