# Peer review of "VISTA, PDL-L1, and BRAF—A Review of New and Old Markers in the Prognosis of Melanoma"

_medicina, 2022, doi:10.3390/medicina58010074_

Round 1
Reviewer 1 Report
The aim of this paper is to perform a narrative review of the old and new markers relevance on determination of the prognosis of the melanoma.
This issue is very relevant and deserves a deeper insight. However, due to the fact that this is a narrative review, and in most of the topics analyzed, the original studies present contradictory results, the authors have to improve the article writing so that there is a flow of intelligent, coherent and understandable ideas in the manuscript. In the present form, it is hard to get a final message from this paper. Additionally, and with critical importance, the authors are not citing the references appropriately: some sentences are not supported by the studies indicates and in other paragraphs, with opposite ideas, all the references are presented at the end of the paragraph, which do not allow the reader to know which studies arrive to each of the conclusions. Examples of this situations are provided below.
Another critical issue is the fact that abbreviatures are used without previous full description of their meaning, and this must be corrected.
Some specific concerns are:
- At second paragraph, it is important to specify other possible locations of the melanoma to present an overview of the problem, before focusing on the topic of this review about cutaneous melanoma;
- L65, more detail about “secondary determinations” is needed to allow proper comprehension of this statement;
- L69, AJCC, before the use of an abbreviature the full designation needs to be presented;
- Breslow index needs to be briefly described. Melanoma is with a typo;
- Description of the nodular melanoma is poor and needs improvement;
- L93 TILs, before the use of an abbreviature the full designation needs to be presented;
- L98 and L 102, Rao et al and Eriksson et al. studies need to be cited after the statements of their observations/results;
- L108 “Regarding sentinel nodules, the TILs score is considered to be inversely proportional to the risk of metastasis when using classifications described by both Clark and the Australian Melanoma Institute.“ this statement needs a reference;
- Immunohistochimestry – typo;
- IHC abbreviature most be presented after the full expression;
- L169, this sentence needs a reference;
- L190 – Repetition of the same sentence-”These findings are extremely important because they shows that tumoral cells and 190 not only inflammatory cells can express this marker [18,20,21].“;
- The paragraph that ends in L202 needs a reference.
Author Response
Dear reviewer,
Thank you so much for your feedback! We will take your concerns and try to explain the changes one by one in the following paragraphs:
Point 1: "the authors have to improve the article writing so that there is a flow of intelligent, coherent and understandable ideas in the manuscript. In the present form, it is hard to get a final message from this paper."- There were multiple structures we thought of while writting the paper but eventually, because there are many targets we tried to approach, we reached the conclusion that each topic should be treated separately. We will try to organise the sections better.
Point 2: "Additionally, and with critical importance, the authors are not citing the references appropriately: some sentences are not supported by the studies indicates and in other paragraphs, with opposite ideas, all the references are presented at the end of the paragraph, which do not allow the reader to know which studies arrive to each of the conclusions"- We added the missing references as follows in the next points you mentioned.
Point 3: "Another critical issue is the fact that abbreviatures are used without previous full description of their meaning, and this must be corrected"- we corrected this mistake by adding the terms in the text.
Point 4: "At second paragraph, it is important to specify other possible locations of the melanoma to present an overview of the problem, before focusing on the topic of this review about cutaneous melanoma"-added into text the mentions for other types of melanoma, introducing the types of mucosal melanoma, ocular melanoma, acral melanoma (cutaneous/sun-shielded skin)
Point 5: "L65, more detail about “secondary determinations” is needed to allow proper comprehension of this statement"- we changed the structure of the sentence to make it more clear
Point 6: "L69, AJCC, before the use of an abbreviature the full designation needs to be presented"- we corrected the mistake and added the full term
Point 7: "Breslow index needs to be briefly described. Melanoma is with a typo", we added the description for the Breslow index and corrected the typo
Point 8: "Description of the nodular melanoma is poor and needs improvement" we added a brief description of nodular melanoma
Point 9: "L93 TILs, before the use of an abbreviature the full designation needs to be presented"- we corrected the mistake and added the full term
Point 10: "L98 and L 102, Rao et al and Eriksson et al. studies need to be cited after the statements of their observations/results" - we added the citations required
Point 11: "L108 “Regarding sentinel nodules, the TILs score is considered to be inversely proportional to the risk of metastasis when using classifications described by both Clark and the Australian Melanoma Institute.“ this statement needs a reference"- we added the required reference
Point 12: "Immunohistochimestry – typo"- we corrected the mistake
Point 13: "IHC abbreviature most be presented after the full expression"- we added the full term in the text
Point 14: "L169, this sentence needs a reference"-we added the required reference
Point 15: "L190 – Repetition of the same sentence-”These findings are extremely important because they shows that tumoral cells and 190 not only inflammatory cells can express this marker [18,20,21].“- we corrected the repetition
Point 16: "The paragraph that ends in L202 needs a reference" -we inserted the required reference.
We appreciate your feedback and I hope we were able to answer your concerns and correct our mistakes.
With respect,
Dr. Andreea Tinca
Reviewer 2 Report
I recommend to explain the difference between brisk and non-brisk infiltrate and also introduce the term Tumor infiltrating lymphocytes before the abbreviation TILs.
Author Response
Dear reviewer,
Thank you so much for your feedback and suggestions!
Point 1: " I recommend to explain the difference between brisk and non-brisk infiltrate" - thank you, I added brief explanations for both topics.
Point 2: "introduce the term Tumor infiltrating lymphocytes before the abbreviation TILs"-thank you, I corrected and added the term.
With respect and regards,
Dr. Andreea Tinca
Round 2
Reviewer 1 Report
Authors performed some of the corrections suggested, however, there are still some important issues to be solved. A detailed double-check of the references presented is needed. It seems that new references were added to this version of the manuscript, and the references from the previous version were not updated. Numerous incongruencies are present which need to be addressed. Please find below a list of examples, not exhaustive:
1 - P3L112, typo error “ver”;
2 - P3L121, reference to Rao et al should be placed at the end of the sentence “patients who did not show inflammatory infiltrate” and only the reference to the study by Eriksson et al should appear after “independent prognostic factor”;
3 - P3L130, please double-check if the reference 10 is sufficient to support this paragraph or if reference 11 should also be included;
4 - P3L132, “A study conducted by Bugovnic, Kakavand and Wong has shown that 132 the presence of TILs in metastatic tumors (inflammatory cells being identified using im- 133 munohistochemistry techniques) are independent prognostic factors [9]” the reference presented is not in accordance with the sentence;
5 - P3L148, This sentence also needs to cite the original paper “A study conducted by Dinehart showed that majority (95%) of surveyed dermatopathologist are using immunohistochemistry (IHC) in their practice”;
6 - P4L187,” One of these studies, led by Resenbaum and Aplin, that is conducted on patients with stage III cutaneous melanoma explains that intratumoral inflammatory infiltrate, also known as TILs, is important for the better prognosis of patients. Thus, the cases evaluated in that study were reported as having either increased or low expression of VISTA and were correspondingly evaluated in terms of 191 survival. [18]. “, in the reference list provided, this study is number 20, not 18;
7 - P6L234 “Combinations of PD1/PD-L1 inhibitors with targeted therapies and other inhibitors of the immune system are being researched. “, this sentence needs a reference and do not justify a paragraph by itself;
8 - P6L295, authors make reference to a meta-analysis “A meta-analysis targeting more than 3400 patients with advanced forms of cancer 295 showed a lower risk of adverse effects in patients” and finish with 2 references. Please cite just the study of the meta-analysis itself.
Author Response
Dear reviewer
Thank you so much for your time and patience! We looked through the entire bibliography and changed the data in our article. Further, we will adress the specific point you brought in our attention:
1 - P3L112, typo error “ver”: checked and corrected
2 - P3L121, reference to Rao et al should be placed at the end of the sentence “patients who did not show inflammatory infiltrate” and only the reference to the study by Eriksson et al should appear after “independent prognostic factor”: mistakes corrected
3 - P3L130, please double-check if the reference 10 is sufficient to support this paragraph or if reference 11 should also be included: refference 10 contains the information required but we added 11 for further back up
4 - P3L132, “A study conducted by Bugovnic, Kakavand and Wong has shown that 132 the presence of TILs in metastatic tumors (inflammatory cells being identified using im- 133 munohistochemistry techniques) are independent prognostic factors [9]” the reference presented is not in accordance with the sentence: we added the particular bibliographies
5 - P3L148, This sentence also needs to cite the original paper “A study conducted by Dinehart showed that majority (95%) of surveyed dermatopathologist are using immunohistochemistry (IHC) in their practice”-we added the targeted study
6 - P4L187,” One of these studies, led by Resenbaum and Aplin, that is conducted on patients with stage III cutaneous melanoma explains that intratumoral inflammatory infiltrate, also known as TILs, is important for the better prognosis of patients. Thus, the cases evaluated in that study were reported as having either increased or low expression of VISTA and were correspondingly evaluated in terms of 191 survival. [18]. “, in the reference list provided, this study is number 20, not 18;-mistake corrected
7 - P6L234 “Combinations of PD1/PD-L1 inhibitors with targeted therapies and other inhibitors of the immune system are being researched. “, this sentence needs a reference and do not justify a paragraph by itself;-reference added
8 - P6L295, authors make reference to a meta-analysis “A meta-analysis targeting more than 3400 patients with advanced forms of cancer 295 showed a lower risk of adverse effects in patients” and finish with 2 references. Please cite just the study of the meta-analysis itself.-reference added
For proper English langage we requested help from the MDPI editors, which analysed and corrected our article. We revised the bibliography and added the missing articles. We will upload a version in which you can easily track the changes made.